# Hepatitis B Virus Genotypes and Subgenotypes Circulating in Infected Residents in a Country with High Vaccination Rate

**DOI:** 10.3390/v16060954

**Published:** 2024-06-13

**Authors:** Carolina Silva, Diogo Ramos, Miriam Quina, Elizabeth Pádua

**Affiliations:** Reference Laboratory of HIV and Hepatitis B and C, Infectious Diseases Department, National Institute of Health, Av. Padre Cruz, P-1649-016 Lisbon, Portugal

**Keywords:** HBV, genotypes/subgenotypes, immigrants, geographic origin, Portugal

## Abstract

Despite the availability of a vaccine against hepatitis B virus (HBV), this infection still causes public health problems, particularly in susceptible populations. In Portugal, universal free vaccination started in 1994, and most HBV infections are diagnosed in immigrants from high-prevalence countries. Our aim was to assess the pattern of HBV genotypes/subgenotypes in samples collected between 2017 and 2021 from a convenience sample of 70 infected residents in Portugal. The HBV pol/HBsAg region was amplified and sequenced, allowing the analysis of RT sequences submitted to phylogenetic analysis and mutations assessment. A total of 37.1% of samples were from native Portuguese, aged 25–53 years (mean: 36.7 years), and the remaining samples were from individuals born outside of Portugal. A high diversity of HBV was identified: subgenotypes A1–A3 in 41.0% (16/39); D1, D3, and D4 in 30.7% (12/39); E in 23.1% (9/39); and F4 in 2.6% (1/39). Besides genotypes A and D, Portuguese were also infected with genotypes E and F, which are prevalent in Africa and South America, respectively. Resistance mutations in RT sequences were not found. The findings provide valuable insights for updating the HBV molecular epidemiology in Portugal. However, successful strategies to prevent and control the infection are still needed in the country, especially among susceptible and vulnerable populations.

## 1. Introduction

Hepatitis B virus (HBV) is a DNA virus belonging to the *Hepadnaviridae* family and the *Orthohepadnavirus* genus. It is the etiological agent of acute and chronic hepatitis B infection with high prevalence in East Asia, Sub-Saharan Africa, and the Amazon region [1,2,3,4]. Despite an effective vaccine to prevent HBV infection, about 1.5 million people become newly infected each year [5]. The World Health Organization (WHO) estimated that around 296 million people were living with chronic hepatitis B infection in 2019, leading to 820,000 fatalities caused by liver diseases [5,6]. In fact, the preventive vaccination programs, diagnostic facilities, and treatment access vary worldwide. Only 10% of the HBV-infected individuals are aware of its status and life-threatening complications; for example, decompensated cirrhosis or hepatocellular carcinoma (HCC) cannot be prevented in 20–30% of chronic carriers [6,7,8].

In Portugal, universal free vaccination against HBV started in 1994 in adolescents (10–13 years old) [7]. Following the WHO recommendations, changes occurred in 2000, and the vaccine began to be administered at birth [7,8]. Vaccination coverage of 94% was obtained in 2010 in newborns, infants, and adolescents. In 2018, coverage reached 98% at the age of 12 months [7]. The national serological survey in vaccine-preventable diseases conducted in 2016 indicated that 72.4% of Portuguese aged over 45 years old were susceptible to HBV [9].

The nucleotide diversity level among HBV genomic sequences led to its classification into ten genotypes (A–J) [10,11]. Globally, genotype A circulates in Europe and North America, and genotypes B and C in Southeast Asia and Southern and Central America [12,13,14]. Genotype D is the most widely distributed worldwide. Genotype E is originally from the African continent [15] and genotype F is native to Central and South America, as well as Alaska [2]. However, some HBV subgenotypes have a distinct pattern of geographical distribution [16,17]. The subgenotype A1 prevails in Africa, whereas A2 is prevalent outside of Africa. The subgenotypes D2 and A2 are widely detected in Europe [10,18].

Recent studies showed an increasing prevalence and incidence of HBV infection and a spread of the different genotypes in European countries. High prevalence rates of hepatitis B surface antigen (HBsAg) were detected in migrants and refugees from outside of Europe [11,19,20]. In France and Italy, the prevalence rates of hepatitis B chronic carriers are higher among people from Sub-Saharan Africa and in people with low socio-economic income [21,22]. In the Netherlands and the United Kingdom, an increase in mortality and morbidity from HCC has been observed among ethnic immigrants living in those countries [23,24].

In 2016, a national survey implemented in the general population in Portugal revealed a low endemicity for HBV infection markers, in contrast to high-risk groups [25]. However, the number of immigrants in Portugal has increased steadily in last years, particularly immigrants coming from Sub-Saharan Africa, Asia, and South America, who accounted for 11.6% of the country’s native population in 2022 [26]. Despite a few studies conducted in public Portuguese hospitals [27,28,29,30], the molecular surveillance pattern of HBV subgenotypes and the increased population heterogeneity regarding the geographic origin of residents have not been updated. The most recent molecular data considering subgenotypes is related to an analysis of samples collected until 2012 [31]. Thus, the main goal of the current study was to perform a new assessment of the diversity of HBV infection by analyzing samples obtained between 2017 and 2021 from residents with chronic hepatitis B infection to update the HBV genotypes/subgenotypes that are circulating in Portugal.

## 2. Materials and Methods

The samples analyzed at the Reference Laboratory of the National Institute of Health (INSA) were collected between 2017 and 2021 from individuals with a diagnosis of chronic hepatitis B infection (persistence of HBsAg for more than six months). Along with these 70 samples received, limited epidemiological information was available to characterize the population (sex, age, and geographic origin of individuals).

All samples were submitted to HBV DNA quantification using the commercial assay COBAS^®^ AmpliPrep/COBAS^®^ TaqMan^®^ HBV, v2.0, following the manufacturer’s recommendations. The lower limit of quantitation (LoQ) of the COBAS^®^ AmpliPrep/COBAS^®^ TaqMan^®^ HBV test v2.0 is 20 IU/mL HBV DNA (Log10 = 1.30), and the test provides a linear response from 20 IU/mL HBV DNA (Log10 = 1.30) to 1.7 × 10^8^ IU/mL HBV DNA (Log10 = 8.23) in both EDTA plasma and serum. Those samples with detectable viral load were selected to assess the HBV genotype/subgenotypes.

HBV DNA was extracted from plasma with the QIAamp^®^ DNA Mini Kit (Qiagen, Hilden, Germany) and submitted to nested PCR by using two sets of primers for amplification of the P/HBsAg coding region, as described in Coffin et al., 2015 and Osiowy et al., 2011 for the first and second round of amplification, respectively [32,33] (Table 1).

For the first PCR round, illustra puReTaq Ready-To-Go PCR Beads (GE Healthcare, Buckinghamshire, UK) was used with the HBVpolyDF and HBVpolyDR primers at a concentration of 0.4 µM, and followed by the addition of 10 µL of the extracted HBV DNA sample in the reaction mixture. The amplification conditions were as follows: 95.0 °C for 5 min, 5 cycles of 95.0 °C for 30 s, 50.0 °C for 30 s and 72 °C for 45 s, and 35 cycles of 95.0 °C for 30 s, 50.0 °C for 30 s, and 72 °C for 45 s, with a final extension of 72 °C for 7 min. For the second PCR round, AmpliTaq Gold DNA polymerase was used with the primers HBVsF01in and HBVR743as at a concentration of 0.4 µM, and 2 µL of the product from the first PCR round was added to the reaction mixture. Amplification conditions were as follows: 95.0 °C for 10 min, followed by 35 cycles of 94.0 °C for 30 s, 55.0 °C for 30 s and 72 °C for 40 s, and a final extension of 72 °C for 7 min.

PCR products were purified with the ExoSap-IT Clean-UP Kit (GE Healthcare, Buckinghamshire, UK) and the sequencing reaction was carried out with the BigDye™ Terminator v1.1 Cycle Sequencing Kit (Applied Biosystems/ThermoFisher Scientific, Waltham, MA, USA) with the PCR internal primers and following the manufacturer’s recommendations. Samples were run through the ABI Prism 3130xl genetic analyzer automatic sequencer (Applied Biosystems, Waltham, MA, USA).

Sequences obtained using Sanger sequencing were analyzed to correct nucleotide misreads and aligned with ClustalW to allow the construction of a partial S consensus sequence for each sample. Phylogenetic analysis was performed through the alignment of the obtained sequences with reference sequences of different genotypes/subgenotypes of HBV retrieved from the GenBank database, using MEGA (Molecular Evolutionary Genetic Analysis) v11.0.9 [34]. The subgenotype reference sequences used were selected based on the classification proposed in Kramvis, 2014 and Yin et al., 2019 [11,35]. The neighbor-joining method and the Kimura two-parameter model were used to generate the tree. The branches’ robustness was assessed using the bootstrap method with 1000 replicates and the clusters were considered significant with values greater than 70%. All ambiguous positions for each sequence pair were removed. Additionally, BioEdit v7.2.5 [36] was used to perform an alignment of the amino acid sequences comprising the RT region of the HBV genome. Positions 80, 169, 173, 180, 181, 184, 194, 202, 204, 236, and 250 were investigated for amino acid substitutions linked to hepatitis B treatment resistance by comparison with the reference sequence NC_003977.2.

This study was approved by the Ethical Committee of the National Institute of Health, Portugal.

## 3. Results

### 3.1. Epidemiological Characterization of the Samples Studied

The set of 70 samples analyzed in the current study correspond to individuals aged between 22 and 57 years old (mean age of 38.6 years). Sixty-seven (95.7%) samples were obtained from males and three (4.3%) samples were obtained from females. Regarding the geographic origin, 26 (37.1%) samples were collected from individuals born in Portugal, while the remaining 44 (62.9%) samples were obtained from individuals born elsewhere: 7 (10.0%) in Eastern European countries, 34 (48.6%) in Africa, and the remaining 3 (4.3%) in South America. The data related to the geographic origin of individuals are described in Table 2.

The mean age among the group of 26 samples corresponding to Portuguese natives was 38.9 (25 to 53) years old, and that for the group of 44 samples from individuals born outside of Portugal was 38.5 (22 to 57) years old.

### 3.2. HBV DNA Quantification

HBV DNA was quantified in 55.7% (39/70) of samples, with values ranging from 119 to 62,500,000 IU/mL (mean of 1,620,525 IU/mL). According to the obtained viral load values, the distribution of samples in the defined ranges was determined: 15 (38.5%) had viral load values of less than 1000 IU/mL (mean of 388 IU/mL, ranging from 119 to 854), 15 (38.5%) had values between 1000 and 10,000 IU/mL (mean of 2,886 IU/mL, ranging from 1110 to 8400), and 9 (23.1%) had values greater than 10,000 IU/mL (mean of 7,016,822 IU/mL, ranging from 11,500 to 62,500,000) (Table 3).

The quantification results of HBV DNA for the remaining 44.3% (31/70) of samples were less than 20 IU/mL (lower limit of quantitation of the assay). Consequently, these samples were excluded from the subsequent molecular analysis.

### 3.3. Classification of HBV Sequences in Viral Subgenotypes

The phylogenetic classification of the 39 P/HBsAg sequences obtained from samples with detectable HBV viral load, which were previously amplified and sequenced, showed that 17 (43.6%) belong to the genotype A cluster, 12 (30.8%) to genotype D, 9 (23.1%) to genotype E, and 1 (2.6%) to genotype F. With the exception of the B38-RT sequence of genotype A, the sequences were distributed along the tree in different subclusters with significant bootstrap values (above 70%), which allowed the determination of HBV subgenotypes (Figure 1).

The distribution of the sequences under analysis based on the geographic origin of individuals, i.e., 48.7% (19/39) native Portuguese and 51.3% (20/39) individuals born outside of Portugal (Eastern Europe, Africa, and South America), is shown in Table 4.

### 3.4. Analysis of Resistance Mutations in the RT Region

The evaluation of amino acid substitutions in all sequences derived from samples with detectable HBV DNA was performed for the RT region of the HBV genome and no drug resistance mutations were detected (Figure 2).

## 4. Discussion

Strong historic and ongoing social and economic links with several West African countries have contributed to the ethnic and cultural heterogeneity of the population living in Portugal. Nevertheless, an increasing number of immigrants coming from South America and Asia, as well as from Eastern Europe, observed in recent years, may contribute to changes in the molecular pattern of infections and raise the diversity of HBV genotypes [13]. In most of these countries, the prevalence of HBV infection remains higher, the diagnosis and treatment are poorly implemented, and the rate of vaccination coverage is low in the population. Currently, guidelines for screening, vaccination, and treatment vary with the available resources in each country [37]. Moreover, it has been shown that HBV-vaccinated individuals respond unequally to different viral subgenotypes and HBsAg subdeterminants [10,38,39]. Thus, the surveillance of HBV subgenotypes and the clinical outcome of this infection is still a matter of particular interest in Portugal.

The aims of the current study were to detect HBV DNA in a convenience sample collected from chronic HBV-infected individuals, in order to update the molecular epidemiology of the circulating HBV subgenotypes in the infected population residing in Portugal. As expected, the majority of samples analyzed in the current study were related to individuals born in West Africa, which corresponded to 48.6% (34/70). In a 2023 estimate from Trickey et al., high values of HBsAg prevalence among migrants were detected in Portugal [40]. Interestingly, 37.1% (26/70) of the samples were collected from native Portuguese infected with hepatitis B, with ages ranging between 25 and 53 years old (mean of 36.7 years). According to this age range, differences in hepatitis B vaccine administration status are expected. In fact, some Portuguese individuals were already adults in 1994, when the national vaccination program was started to cover teenagers. Unfortunately, no data from HBV vaccination status of the infected individuals were available for further discussion, although a proportion of Portuguese individuals would be eligible for vaccination against hepatitis B when they reached adolescence [7], particularly those with ages below 33 years old. Furthermore, it is recognized that the administration of the hepatitis B vaccine to adolescents between 1994 and 1999 could lead to failures in completing the vaccination schedule, which comprises three doses of vaccine over a period of 6 months. In line with WHO recommendations, the change in the age for HBV vaccine administration made from 2000 onwards in Portugal can reduce the incidence of incomplete vaccination schedules and increase vaccination efficacy [7,8]. However, in the current study, no individuals would have been covered by this change, which may explain the results obtained.

In African countries, hepatitis B vaccination coverage is low [41] and most HBV infections occur early in life, mainly by vertical transmission and by exposure to blood or other contaminated body fluids. In contrast, low rates of HBV infection in European countries have been reported, with the main routes of transmission being sexual contact and needle sharing [42]. Furthermore, a higher HBV viral load is often associated with an increased progression of the liver disease and development of HCC [12,14,43,44]. HBV DNA was undetectable in 44.3% (31/70) of the analyzed samples, which suggests that these chronically infected patients may be under treatment for HBV infection. However, no data regarding HBV treatment are available, preventing a more detailed discussion of the results. Therefore, these samples were excluded from further molecular analysis. For the remaining 55.7% (39/70) of samples, the DNA viral load was quantified. The association of viral load values with the demographic origin of individuals revealed that 48.7% (19/39) were Portuguese natives and 38.5% (15/39) were derived from immigrants from West Africa. Most of these samples showed low DNA viral loads (between 119 and 854 UI/mL). However, 12.9% (9/39) of the samples analyzed showed high DNA viral loads (between 10,500 and 62,500,000 IU/mL). Additionally, the analysis of RT amino acid sequences showed no resistance mutations in the consensus sequences obtained. Once again, the lack of information regarding the absence of HBV treatment and/or the time of HBV acquisition for these individuals did not allow any further discussion of results.

HBV genotypes obtained in the study agreed with the data previously gathered from blood donors and patients attending Portuguese public hospitals, which showed a prevalence of infections caused by genotypes A and D [27,28,31]. The results of our study are in agreement with HBV genotyping results obtained in previous studies, which indicate a higher prevalence of genotypes A and D, compared to genotype E. Nevertheless, a higher genotype diversity of HBV was identified, namely the presence of the subgenotypes A1 to A3 in 41.0% (16/39) of the samples, and D1, D3, and D4 in 30.7% (12/39) of the samples, followed by genotype E and F4 in 23.1% (9/39) and 2.6% (1/39) of the samples, respectively. Besides genotype A and D infections, the native Portuguese were also infected with genotypes E and F, which were originally prevalent in Africa and South America, respectively. However, both of these genotypes have been identified in European immigrants or travelers who were infected abroad [30,31]. Marcelino et al., 2022, showed a higher prevalence of infections with genotypes A, D, and E, but no infections with genotype F were identified in the studied samples, which were collected until 2012 [31]. To the best of our knowledge, our study described, for the first time, the presence of the HBV subgenotype F4 in a Portuguese native. In the current study, infections with genotype E were also found in samples from residents born in Cape Verde, Guinea-Bissau, Angola, and S. Tome and Principe, and in one Portuguese native.

With the exception of A3 and D1 infections, all subgenotypes assessed in the current study were identified in Portuguese natives (A1, A2, D3, D4, E, and F4). According to the literature, the HBV subgenotypes A1, A4, and A3 are found mainly in Africa, whereas A2 was found in Europe and North of America [19]. Additionally, subgenotypes A2 and D1 were related to infected individuals born in Romania and Georgia. These findings corroborate previous research indicating a high prevalence of these two subgenotypes in Europe. Despite the high similarity (>95%) of most sequences derived from the analyzed samples, no duplicated sequences were found. This high similarity was expected since HBV is a DNA virus with several degrees of adaptation to human populations [45].

## 5. Conclusions

Due to the recent large-scale immigration to Europe, a change in the molecular epidemiology of HBV chronic infection and the burden of disease in the upcoming years may be an expected scenario. Despite the limitation of the sample size and lack of epidemiological and clinical data, our findings regarding viral diversity assessed in the current study provide valuable insights into the molecular epidemiology of HBV infection in Portugal. The results obtained in our study agreed with previously published data. However, it is important to point out the presence of the HBV subgenotype F4 in the country, which was not previously reported. To the best of our knowledge, our work, together with that of Marcelino et al., 2022 [31], is among the first to evaluate HBV subgenotypes in Portugal. Any successful strategy to prevent and control HBV infection requires effective strategies for testing and implementation of vaccination programs against HBV, especially among the susceptible and most vulnerable populations.

## Figures and Tables

**Figure 1 viruses-16-00954-f001:**
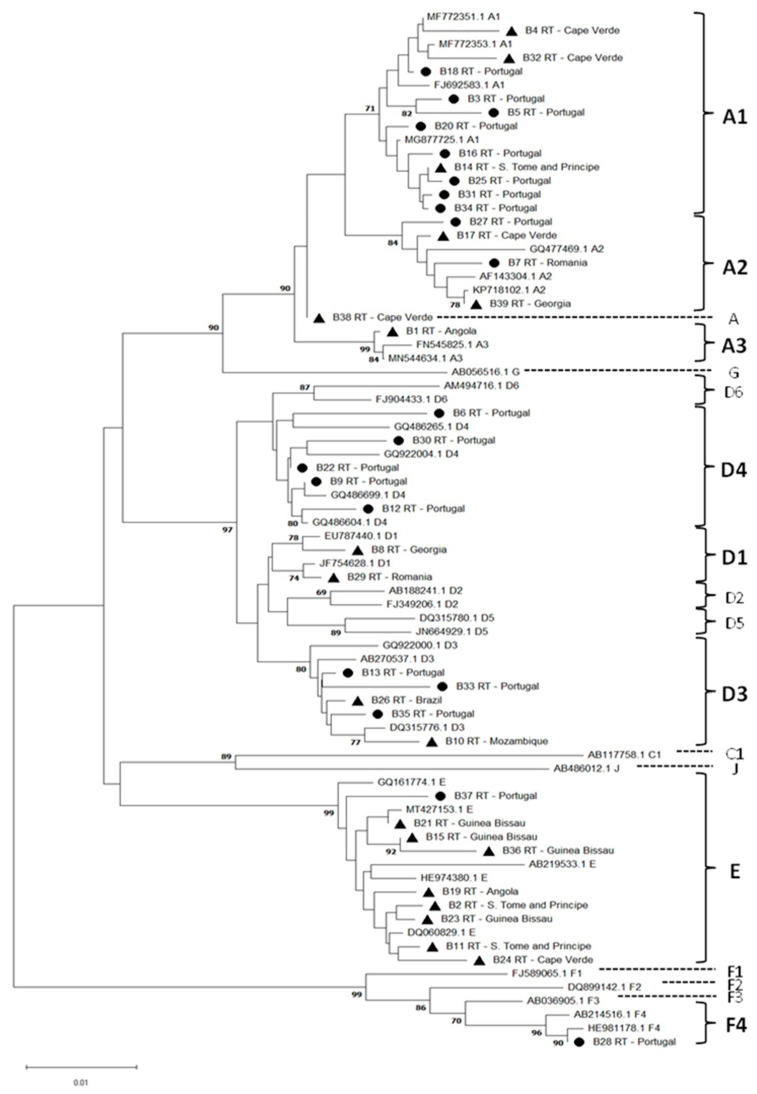
Phylogenetic neighbor-joining tree of HBV sequences based on the alignment of 733 nucleotides of the P/HBsAg region along with 37 reference sequences from HBV genotypes and subgenotypes. The sequences under analysis are highlighted: native Portuguese with ● and individuals born outside of Portugal with ▲. Reference sequences are represented by the accession number and HBV subgenotype. The scale bar indicates the number of substitutions per site.

**Figure 2 viruses-16-00954-f002:**
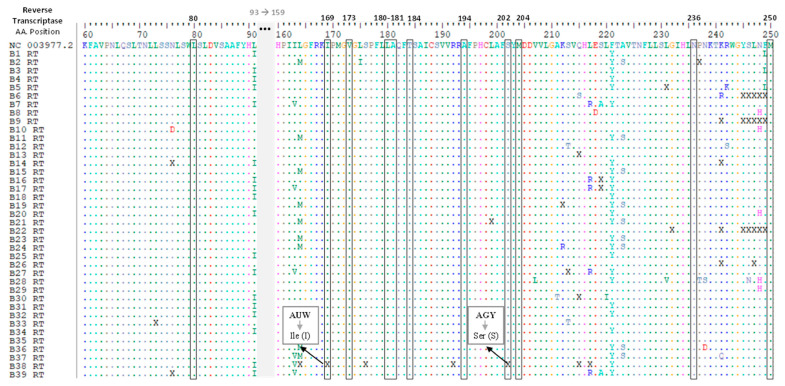
Amino acid alignment of the sequences under study for the reverse transcriptase (RT) region of the HBV genome. The first line contains the reference sequence used for the drug resistance testing. The analyzed positions are delimited by rectangles. The amino acids corresponding to triplets where one nucleotide could not be determined are marked with an “X” and all amino acid coding hypotheses were evaluated.

**Table 1 viruses-16-00954-t001:** Oligonucleotide primers used for the nested-PCR amplification and sequencing of the genomic region of HBV.

Primer Name	Usage	Direction	Sequence (5′→3′)	Location ^1^	Fragment Size
HBVpolyDF	1st PCR round	Sense	CCTGCTGGTGGCTCCAGTTCAG	58–79	1140
HBVpolyDR		Antisense	GTTGCGTCAGCAAACACTTGGC	1177–1198	
HBVsF01in	2nd PCR round and sequencing	Sense	ACCCTGYRCCGAACATGGA	143–161	758
HBVR743as		Antisense	CAACTCCCAATTACATARCCCA	880–901	

^1^ Numbering according to the reference sequence NC_003977.2.

**Table 2 viruses-16-00954-t002:** Distribution of the samples analyzed according to the geographic origin of the individuals.

Samples Characterization (N = 70)			
Region/Country of Birth		N	%
Portugal		26	37.1
Eastern Europe		7	10.0
	Moldova	2	2.9
	Georgia	2	2.9
	Romania	2	2.9
	Ukraine	1	1.4
Africa		34	48.6
	Angola	14	20.0
	Cape Verde	8	11.4
	Guinea-Bissau	6	8.6
	Mozambique	5	7.1
	S. Tome and Principe	1	1.4
South America		3	4.3
	Argentina	2	2.9
	Brazil	1	1.4

**Table 3 viruses-16-00954-t003:** Distribution of the samples according to geographic origin and the HBV DNA quantification (intervals range) results obtained.

Region/Country of Birth		HBV DNA Quantification (N = 70) (Interval in IU/mL)			
		<20 ^1^	[20–1000]	[1000–10,000]	>10,000
Portugal		7	6	9	4
Eastern Europe	Georgia	-	1	1	-
	Romania	-	1	1	-
	Moldova	2	-	-	-
	Ukraine	1	-	-	-
	Subtotal	3	2	2	-
Africa	Cape Verde	9	1	3	1
	Guinea-Bissau	4	3	1	-
	S. Tome and Principe	3	-	-	3
	Angola	3	1	-	1
	Mozambique	-	1	-	-
	Subtotal	19	6	4	5
South America	Brazil	1	1	-	-
	Argentina	1	-	-	-
	Subtotal	2	1	-	-
Total (%)		31 (44.3%)	15 (21.4%)	15 (21.4%)	9 (12.9%)

^1^ Undetectable HBV DNA.

**Table 4 viruses-16-00954-t004:** HBV genotypes and subgenotypes assessed in the samples analyzed from native Portuguese and individuals born outside of Portugal (Eastern Europe, Africa, and South America).

Country of Birth	HBV Genotype/Subgenotype (N = 39)									Total N (%)
	A ^1^	A1	A2	A3	D1	D3	D4	E	F4	
Portugal	-	8	1	-	-	3	5	1	1	19 (48.7%)
Eastern Europe	-	-	2	-	2	-	-	-	-	4 (10.3%)
Georgia	-	-	1	-	1	-	-	-	-	2
Romania	-	-	1	-	1	-	-	-	-	2
Africa	1	3	1	1	-	1	-	8	-	15 (38.5%)
Cape Verde	1	2	1	-	-	-	-	1	-	5
Guinea-Bissau	-	-	-	-	-	-	-	4	-	4
S. Tome and Principe	-	1	-	-	-	-	-	2	-	3
Angola	-	-	-	1	-	-	-	1	-	2
Mozambique	-	-	-	-	-	1	-	-	-	1
South America	-	-	-	-	-	1	-	-	-	1 (2.6%)
Brazil	-	-	-	-	-	1	-	-	-	1
Total N (%)	1 2.6%	11 28.2%	4 10.3%	1 2.6%	2 5.1%	5 12.8%	5 12.8%	9 23.1%	1 2.6%	39

^1^ Sequence with subgenotype not determined.

## Data Availability

HBV sequences obtained in the current study were submitted to the GenBank database under the accession numbers PP417850 to PP417888.

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
