# Peer review of "Hepatitis B Virus Genotypes and Subgenotypes Circulating in Infected Residents in a Country with High Vaccination Rate"

_viruses, 2024, doi:10.3390/v16060954_

Round 1

Reviewer 1 Report

Comments and Suggestions for Authors

Dear Authors,

the manuscript is well-written and contains novel and valuable results.

However, I have some comments and suggestions that can help to improve the overall readability of the manuscript.

1. Please, provide more detailed description of Methods used to align sequences and carry out phylogenetic analysis. It is significant for evaluating the results provided.

2. Were there some duplicated or highly similar (with similarity over 95%) sequences after sequencing and bioinformatics analysis and determined in the samples obtained from various people. Please, discuss it in the text of the manuscript.

3. What were the methods for evaluating prevalent variants of viruses extracted from the samples obtained from patients with HBV infection. In general please point out mean value and range of viral load of patients included in the study.

4. I suggest that you describe in detail the inclusion and exclusion criteria for the participants of the study.

 5. Please, compare the results obtained in the study with earlier published findings and emphasize the novelty of the research.

Comments on the Quality of English Language

Please re-phrase the Title to shorten it, if possible, and check English with the help of a native speaker.

Reviewer 2 Report

Comments and Suggestions for Authors

General comment

The paper is well within the scope of Viruses. The pattern of HBV subgenotype prevalence is continuously changing and particularly interesting in a country with a large variety of immigrants or re-immigrants over many decades like Portugal. The presentation of the data is clear, but there are many weaknesses:

       I.          The number of the studied participants with CHB is small: they begin with only 70, but only 39 had detectable HBV DNA. Only 3 of the 70 were female. This cannot be representative for the HBV positive people in Portugal. Many data on the ethnic background or the history of vaccination or clinical hepatitis are not available.

      II.          The diagnostical methods are incompletely described and many important HBV markers like HBeAg or antiHBc are not available.

    III.          The authors mention the HB vaccination in the title, but they do not analyze the HBsAg amino acid sequence for the HBsAg subtypes and vaccine escape mutations although they have the HBsAg gene sequence.   

Specific points

1.      The title is fine, but „Update on” could be deleted, because “circulating” implies the actual situation.

2.      L 15. At least as relevant as the RT sequence is the HBs protein sequence since it may cause vaccine escape.

3.      L 16, 17. More relevant than the place of birth is the ethnicity of the parents. Are such data available?

4.      L31. Ref. 1 is outdated, ref. 2 and 3 are not geographically well selected for Portugal. Only 1 participant with HBV DNA was from Brazil, only one from Mozambique.

5.      L 32 and 76. Chronic hepatitis B or chronic HBV infection?

6.      L 52. According to the new data in the paper, the subgenotype prevalence is in Portugal obviously not the same as in entire or in northern Europe: there were only 1/19 with A2 and 0/19 with D2. Thus, the sentence should be reworded, because Portugal is a long-time member of Europe.

7.      L 75. For what does INSA stand?

8.      L 76. Diagnosis of CHB.

a.      What was the limit of detection for HBsAg and the test kit?

b.      Was the specificity of the HBsAg result confirmed?

c.      Was HBeAg tested for?

d.      Why were so few women included? Did one or more contain of them detectable HBV DNA?

9.      L 81 and 137. HBV DNA: What was the limit of detection and/or the limit of quantitation?

10.   L 118. The authors should also analyze the sequence of HBsAg and determine the HBsAg subtypes including the w subdeterminants w1-4 following the schedule from Gerlich et al. Peculiarities in the designations of hepatitis B virus genes, their products, and their antigenic specificities: a potential source of misunderstandings. Virus Genes. 2020 Apr;56(2):109-119. doi: 10.1007/s11262-020-01733-9. PMID: 32026198

a.      Furthermore, potential vaccine escape mutations in the HBsAg sequence should be identified.

11.   L 126. The extreme bias to male participants should be explained here or in the discussion. Are pregnant women not tested for HBsAg in Portugal?

12.   L135. Table 2. I suggest to combine the participants from Eastern Europe in one sub-group although Georgia is politically (not yet?) in Europe.

13.   L 180-182. According to point 10, a figure 3 should be generated with the HBsAg subtype-determining amino acids and/or vaccine escape mutations should be included.

a.      The HBV subgenotypes and the HBsAg subtypes have a strong influence on the full protection by the HB vaccine.

b.      Refs. 10 and 37 do not cover this point sufficiently. HBV subgenotypes, HBsAg subtypes and HB vaccines are better covered in Gerlich WH. Do HBsAg subdeterminants matter for vaccination against hepatitis B? Virus Genes. 2024 Apr;60(2):240-242. doi: 10.1007/s11262-024-02061-y and in the refs there.

14.   References. Why are refs. 6, 24 on HCV included?

Comments on the Quality of English Language

The English is overall ok aside from minor errors.

Reviewer 3 Report

Comments and Suggestions for Authors

Carolina Silva, Diogo Ramos, Miriam Quina, and Elizabeth Pádua present work regarding the molecular epidemiology of a cohort of HBV samples obtained from a reference laboratory from 2017-2021. Proper sample processing and DNA sequencing were performed, finding a high diversity of HBV subgenotypes and genotypes. The authors conclude that monitoring the changing landscape of HBV infection by testing and vaccination is crucial as part of the strategies to control the spread and acquisition of this disease.  

The work is well-performed and nicely written. The data is concise and clearly explained. I have some questions for the authors. 

1) In the Methods section, the authors state that the samples were from subjects with chronic hepatitis B infection. However, it is unclear how they were classified as chronic patients, given that they were HBsAg positive and HBV-DNA negative. I would appreciate it if the authors could clarify this point. 

Regarding the prevalence of the HBV subgenotypes and genotypes, the authors mention that only one past study has been performed on the molecular epidemiology of HBV in Portugal. Are there no other studies? There are no differences in the distribution between the ones reported earlier and this study. Please verify references and clarify.

Comments on the Quality of English Language

Please check the spelling of the word "refuges" on line 56 and any other spelling mistakes that may gone unnoticed. 

Round 2

Reviewer 1 Report

Comments and Suggestions for Authors

Dear Authors,

thank you for correcting the manuscript taking into account my comments and suggestions. I also have a minor comment. Would you please discuss how the results correspond high vaccination rate criterion. As it can be presumed from the manuscript, the samples for sequencing were obtained from people with the age ranged from 22 to 57 years. I suppose that vaccination rate can be different and may vary depending the age range of individuals. Can the authors comment on this and analyse the possible bias that can be associated with my observations. Also, please indicate advantages and limitations of the study.

Comments on the Quality of English Language

Minor English language correction are needed.

Author Response

Reviewer 1

Dear Authors,

thank you for correcting the manuscript taking into account my comments and suggestions. I also have a minor comment. Would you please discuss how the results correspond high vaccination rate criterion. As it can be presumed from the manuscript, the samples for sequencing were obtained from people with the age ranged from 22 to 57 years. I suppose that vaccination rate can be different and may vary depending the age range of individuals. Can the authors comment on this and analyse the possible bias that can be associated with my observations. Also, please indicate advantages and limitations of the study.

[Authors] We appreciate your reviews.

We tried to clarify this matter in the manuscript by describing with more detail the possible bias associated with the age distribution and geographical origin of individuals. We also added more information regarding possible causes of HBV infection in individuals expectedly covered by the vaccination criteria, regarding the national vaccination program.

The advantages of our work include the fact that is one of the very few studies published which described HBV subgenotypes that were identified in samples collected more recently, and according to our knowledge, is the first study that report the infection with F4 in Portugal. The limitations of the study include the small sample size (convenience sample) and the lack of data regarding the HBV treatment and vaccination status of the infected-individuals. These points are covered in the conclusion.

We thank you for providing suggestions and comments that led to the improvement of our manuscript.

Reviewer 2 Report

Comments and Suggestions for Authors

The authors have responded quite well to my suggestions although they could not provide most of the data which would have been useful. Nevertheless, the manuscript has been improved.

Author Response

Reviewer 2

The authors have responded quite well to my suggestions although they could not provide most of the data which would have been useful. Nevertheless, the manuscript has been improved.

[Authors] We completely agree that by providing more information we could further improve the manuscript. Unfortunately, the desired data is not accessible.

We will keep your comments and suggestions in mind when designing the next study. Thank you.